# Supporting Premature Infants’ Oral Feeding in the NICU—A Qualitative Study of Nurses’ Perspectives

**DOI:** 10.3390/children9010016

**Published:** 2021-12-28

**Authors:** Evalotte Mörelius, Charlotte Sahlén Helmer, Maria Hellgren, Siw Alehagen

**Affiliations:** 1School of Nursing and Midwifery, Edith Cowan University, 270 Joondalup Drive, Joondalup, WA 6027, Australia; 2Perth Children’s Hospital, Nedlands, WA 6009, Australia; 3Department of Health, Medicine and Caring Sciences, Division of Nursing Sciences and Reproductive Health, Linköping University, 581 83 Linköping, Sweden; charlotte.sahlen.helmer@liu.se (C.S.H.); maria.hellgren@outlook.com (M.H.); siw.alehagen@liu.se (S.A.)

**Keywords:** breast feeding, human milk, infant, intensive care, lactation, newborn, nurses, parents, premature

## Abstract

One major task in the neonatal intensive care unit (NICU) involves ensuring adequate nutrition and supporting the provision of human milk. The aim of this study was to explore nurses’ experiences of the oral feeding process in the NICU when the infant is born extremely or very preterm. We used a qualitative inductive approach. Nine nurses from three family-centered NICUs were interviewed face-to-face. The interviews were transcribed verbatim and analyzed using content analysis. Five sub-categories and two generic categories formed the main category: ‘A complex and long-lasting collaboration.’ The nurses wished to contribute to the parents’ understanding of the feeding process and their own role as parents in this process. The nurses’ intention was to guide and support parents to be autonomous in this process. They saw the family as a team in which the preterm infant was the leader whose needs and development directed the feeding and the parents’ actions in this process. Written and verbal communication, seeing all family members as important members of a team and early identification of the most vulnerable families to direct the emotional and practical feeding support accordingly can strengthen the feeding process in the NICU.

## 1. Introduction

Nurses working in neonatal intensive care units (NICUs) have large and complex clinical roles that involve caring for preterm babies and sick newborns in need of intensive care. Nurses need specific qualifications to provide time-urgent care for vulnerable infants in high-tech environments [1]. At the same time, they should also provide emotional support to parents who are experiencing mixed feelings of sorrow, uncertainty and happiness [2]. Nurses from Australia have described their work in the NICU as challenging to some extent; work that demands special training, experience and seniority [3].

Working in a family-centered (FC) NICU means that parents are present and involved in the care of the infant around the clock [4,5]. Since a premature infant’s hospital stay can last for several weeks or months [6], close collaborations and relationships may develop between the nurses and the families [7]. Parents can sleep in the same room as the infant and provide most of the care [5]. However, nurses are still responsible for the advanced care of the infant and for providing breastfeeding and lactation support for the mothers of preterm infants; the latter is a delicate task, since the mothers are in such a vulnerable situation [8].

One major task for nurses in the NICU involves ensuring that infants receive adequate nutrition and supporting the provision of human milk [1]. If the parents’ desire is to breastfeed, the nurses should support that long and complex process and encourage early skin-to-skin contact and breastmilk expression [9,10,11]. According to WHO and UNICEF’s tens steps for successful breastfeeding, mothers should be supported to initiate and maintain breastfeeding, manage common difficulties and recognize and respond to their infants’ cues for feeding [9]. For mothers of preterm infants, it is also important to ensure that breastfeeding support is available during the infants’ whole hospital stay [12].

In 2014, nurses in the US reported that they only provided breastfeeding support to 14% of the infants they cared for on a regular weekday shift. The percentage was slightly lower on weeknights (13%) and higher on weekend shifts (18%) [13]. The same study also reported that better staffing was associated with a higher percentage of infants receiving breastfeeding support in the NICU [13]. However, it was up to the nurse to define the meaning of ‘breastfeeding support’ and what to include in the concept when answering the questions [13]. In 2017, another study including nurses (*n* = 71) reported that half of them had provided lactation-based support during their last shift [14]. Lactation support was defined with 21 statements, demonstrating that it can include several different tasks. The most frequently provided lactation support included educational tasks related to breastmilk pumping, breastfeeding, or human milk oral care [14]. The more practical tasks included taking care of the milk for cooling/freezing, defrosting the milk and weighing the child before and after feeding. Some of the less commonly used lactation-support tasks included assisting with skin-to-skin care and the use of a nipple shield [14].

In a recent qualitative study from FC NICUs in Sweden, mothers of extremely preterm infants highlight that the two major reasons for feeling guilt as a mother in the NICU are not being able to produce enough breastmilk and not managing to breastfeed the infant [15]. The mothers described the long process involving milk expression and breastfeeding as a struggle for the infant’s best interests. They found their partners to be the best source of support, but expressed a need for more emotional and practical support from nurses in order to manage to express milk and practice breastfeeding for many weeks [15]. Fathers in FC NICUs have described that they give as much support to the mothers as they can, but they find it difficult to know how best to support mothers when the breastmilk ceases and, in particular, to encourage mothers to pump when there is no milk [16]. The fathers wanted more support from nurses, for the mothers and for themselves [16].

Taken together, families in NICUs need a great deal of oral feeding support from nurses. At the same time, nurses have several families to attend to in an already busy intensive care context. This increases the risk of parents feeling unsupported, which in turn may jeopardize the breastmilk production and further decrease breastfeeding rates for preterm infants. Several studies can be found on barriers and facilitators for successful breastfeeding [11,17,18,19,20], but studies exploring nurses’ experiences of supporting parents with the oral feeding process of their preterm infant in the FC NICU are scarce. It is important to consider nurses’ intentions and views of supporting families in this vulnerable situation. Therefore, the aim of the present study was to explore nurses’ experiences of the oral feeding process in neonatal intensive care units in Sweden when the infant is born extremely or very preterm.

## 2. Materials and Methods

### 2.1. Design

A qualitative inductive approach with face-to-face interviews. Data were analyzed with content analysis according to Elo and Kyngäs [21].

### 2.2. Setting

This study was conducted in three Swedish FC NICUs where parents could stay around the clock with their preterm infants. So that parents can stay and care for their infants at the hospital, temporary parental allowance is available for parents in addition to the 480 days of paid parental leave provided for all parents. Together, the three NICUs have 42 beds. Nurses are responsible for breastfeeding and lactation support and there are no lactation consultants.

### 2.3. Participants

The inclusion criteria were registered nurses working in a NICU with experience of the oral feeding process for extremely and very preterm infants. Nine registered nurses participated. They were all women 27–57 years old (mean 41.5) with 3–30 years (mean 11.8) of work experience of neonatal intensive care. Six of the participants were post-graduate specialists in pediatric and neonatal nursing including a one-year Master of Science degree and three were under-graduate nurses with a Bachelor of Science degree. Six had their own experience of breastfeeding.

### 2.4. Ethics

Consent to conduct the study was sought from heads of pediatric departments and nurse managers at three hospitals. The nurse managers distributed written information about the study to the nurses through the work email distribution lists. Those who expressed an interest in participating received oral and written information about the study and provided written consent to participate at the time of the interview. The study was conducted according to the 2013 Declaration of Helsinki and was approved by the research ethics committee at the appropriate university (2012/162-31).

### 2.5. Data Collection

An interview guide with semi-structured questions was used. The interview dialogue opened with a request to describe a situation when the feeding process went well. The nurses were encouraged to talk about breastfeeding, breast milk expression, nasogastric tube feeding and breastmilk supplements. They were also asked to talk about their view of breastfeeding when the infant was born at different gestational ages and their own role in the infant’s feeding process. Probing questions were asked to deepen and widen their narratives. Demographic data were collected before ending the interview. The interviews were performed in a separate, quiet room at the nurses’ workplaces between December 2014 and February 2015 and lasted from 28 to 66 min (mean 46.4). Data saturation was reached when no new information was obvious during interviews. Two of the authors (CSH, MH) conducted the interviews, which were digitally recorded, de-identified and transcribed verbatim.

### 2.6. Data Analysis

The inductive analysis followed the four steps described by Elo and Kyngäs [21]. First, the interview material was read several times to apprehend its essential features. Second, units of meaning were highlighted and codes were identified. Third, sub-categories were formed from the codes and units of meaning. Fourth, the sub-categories were merged into generic categories and finally into one main category. The analysis was a back-and-forth process between the steps, where all authors were involved in the discussion to reach a final agreement among all authors.

### 2.7. Trustworthiness

Validity was ensured by describing and following a well-used analysis process [21,22]. The interview guide was pilot tested with one nurse to confirm content. The interviews with this specific target group generated rich data [23]. To increase the trustworthiness, the context and participants are described and the findings are supported with quotations from the interviewees named respondents A–I. The analysis was discussed within the research group and the final findings were agreed by all the researchers. The pre-understanding of the studied subject comes from the authors’ professional occupations as midwives, registered nurses and specialists in pediatric and neonatal care. Two of the authors have many years of NICU experience, working with extremely and very preterm infants and their families. All authors are female. To further ensure the trustworthiness of the findings, a data source triangulation was performed in November 2021 [16,24]. Three nurses currently working in the participating NICUs read the final data analysis and provided feedback. One of the nurses was a respondent and the other two were females with five and six years of experience working in neonatal intensive care. All three agreed that the study findings were concordant with current working practice and philosophy. One of the nurses wrote, “… we base our work on the child being a competent individual, we try to see for ourselves and help the parents’ to see the child’s cues in order to establish a feeding situation that works.”

## 3. Findings

The analysis resulted in five sub-categories, two generic categories and one main category (Table 1). The main category derived from the data analysis was ‘A complex and long-lasting collaboration,’ emphasizing how the nurses experienced the feeding process when the infant was born extremely and very preterm. The process was described as unique and vulnerable due to the unexpected early birth, which resulted in aggravating feeding factors for both the mother and the infant. The nurses had an intention to contribute to the parents’ understanding of the feeding process, as this is critical for developing the necessary competence and trust to understand the infant’s development and their own role as parents in this process. The nurses described many kinds of support they gave to the parents with the aim of empowering them. This included practical support, such as how to stimulate the milk production by hand or electric pump and emotional support for parents who experienced distress due to an unexpected situation for which they were unprepared, including a demanding feeding process. The nurses saw the family as a team with all members being involved in the feeding process, which corresponded to what an FC NICU is based on. The preterm infant was described as the leader of the team, as his or her needs and development directed the feeding and the parents being with their child. The nurses supported and guided the family team through this process.

### 3.1. Guiding through the Feeding Process

#### 3.1.1. A Vulnerable Process

The long, extended process—ranging from when the preterm infant was born until when breastfeeding was established—was described as a vulnerable process. This process was strongly connected to the unexpected premature birth and related factors. “I believe… if they get a good start, I believe they benefit from that… In the beginning it is support with pumping and getting to know their infant.” (Respondent G)

Poor medical conditions for the mother, anxiety and worry and low self-efficacy and self-confidence were mentioned as hindering factors for breastfeeding, while calm and confident mothers with rich breastmilk production were considered beneficial. “Right from the beginning, this mother took a very positive attitude toward breastmilk and it felt as if she had already had time to think about breastfeeding, even though the baby was born extremely preterm. She was looking forward to it and was curious about it… so she was very happy and positive… It was positive that she had that curiosity and willingness.” (Respondent H)

Another factor was the infants’ feeding abilities, which were affected by their health status. For instance, the nurses mentioned that forms of breathing support such as ventilators, CPAP and oxygen therapy were obstacles that hindered oral feeding. To overcome these obstacles, the nurses underlined the importance of skin-to-skin contact when establishing breastfeeding. The positive touch from skin-to-skin contact was described as important to balance all the unpleasant care procedures that are necessary while in intensive care and to potentially reduce the risk of feeding problems as the infant grows. The nurses described their efforts to make sure the infants received positive touch even when the parents were not able to be present. Examples of positive touch techniques that nurses could carry out included giving a pacifier to an infant who searched for something to suck on and giving some milk in the mouth while tube feeding.

Further factors that could affect the feeding process were attitudes from society, the family’s extended family and friends and staff. Likewise, nurses’ rosters could hinder the feeding process due to a lack of continuity, requiring the parents to explain repeatedly to staff which phase of the feeding process they were currently in. Moreover, ward routines—like feeding according to the clock, rather than after the infants’ needs—could hinder breastfeeding progress. A positive, calm and safe ward environment was mentioned as something that might strengthen the parents in this vulnerable process.

#### 3.1.2. Contributing to an Awareness of a Different Breastfeeding Process

The nurses had the intention to provide parents with information about the feeding process as soon as possible after the preterm infant was born. The reason for this was to contribute to an awareness of the different and extended timeframe from birth until oral feeding when the infant is born extremely or very preterm compared to late preterm or full-term. “[I] make them understand that everything is not that easy and obvious with breastfeeding. Instead, it can actually be rather difficult and that’s not just for them, it’s in general.” (Respondent E)

The nurses wished to mediate awareness to the parents about preterm infants’ developmental stages and that the feeding should be individualized to follow the infant’s development and maturation. They wanted the parents to know that following the infant’s stages is a matter of patience. “If one delivers a baby in [week] 29, it will take a little longer [to breastfeed]. Maybe the breastfeeding in the beginning is just that the baby is lying by the breast and licking, and maybe trying to suck or just having the nipple in the mouth.” (Respondent F)

They also wanted to increase parents’ understanding of the parental role in the breastfeeding process; how parents could take on more responsibility along the way as they learnt how to read the infant’s cues and behaviors. “They [the parents] often start to watch [the infant] and think, oops he is awake now, or he is sleeping very deeply now, but he should eat. Can I wait?” (Respondent B)

The importance of contributing to an understanding of the unique breastfeeding process was related to the nurses’ experiences of how vulnerable mothers of preterm infants are. They believed that mothers’ insecurity about the feeding process was related to a lack of awareness that infants can be born preterm, a lack of knowledge about preterm infants’ competences and a fear that something bad would happen to the infant. The nurses described how parents seemed to search for knowledge and information about breastfeeding on the internet and by interacting with other parents in similar situations.

The nurses described different ways to talk about the feeding process with the parents. For example, some informed the parents that the infant usually had to reach a certain postnatal week before they could suckle, while others informed them that breastfeeding should be initiated according to the infant’s maturation and abilities. They tried to individualize the information they provided and acknowledged that the information had to change over time. At the same time, they were aware that their personal ways of expressing information to the parents could sometimes be confusing and even misinterpreted by the parents. The current feeding process could be affected by a mother’s previous experience of breastfeeding. If these experiences were linked to negative feelings, the nurses felt that it was extremely important to help the mothers to understand that all feeding processes are unique and that this one could be different to previous feeding experiences.

#### 3.1.3. Providing Professional Support

The nurses described that a positive approach was central when providing support to parents. Giving individual support was perceived as a balancing act so that the feeding process would not be too demanding. They emphasized patience, a positive manner, responsiveness, honesty and a consolatory and encouraging approach as especially important. It was crucial to treat the mothers with empathy and consolation, especially in situations when they indicate distress, for example when breastmilk production was low.

The nurses highlighted the importance of involving the parents in the feeding process and guiding them through the process to empower them. They involved the parents immediately after birth and often the parents could gradually take on more responsibility for the feeding. Involving the parents included guiding them to check the placement of the tube, tube feeding, planning the feeding in collaboration with the nurse and asking the parents about their thoughts and their observations of the infant. Moreover, trusting the parents’ ability to handle and take responsibility for the infants’ feeding was important. “Be nearby, let the parents be close to the baby. As soon as possible, start to involve them in the care [of their infant].” (Respondent G)

The involvement of the parents became more difficult when the infant had continuous tube feeding with a pump, since the parents’ task of tube feeding disappeared when the machine took over and replaced the parents’ actions.

The nurses described how they admired mothers for struggling to keep up breastmilk production for such a long time when an infant was born preterm. Some mothers with no previous intention to breastfeed seemed willing to express breastmilk in order to give that to their preterm infants. They found expressing milk several times a day for months to be an enormous effort on the part of the mothers and they wanted to support them in various ways to persevere. “… but many mothers are extremely… enduring or they, yes they, they are fantastic!” (Respondent A)

In order to support the mothers to produce breastmilk, the nurses encouraged early stimulation by hand or electric pump. They found early stimulation to be important to achieve milk production that met the needs of the infant. However, they were careful to note that it was the mother’s choice whether or not to express milk and that she had the right to make an informed decision. “It is information that we must give the mother so they [the parents] at least know what it is they choose to do if they opt out from breastfeeding and pumping. And maybe discuss it several times because it is not too late just because some days have passed, even if it is better if one starts early.” (Respondent I)

A good start to milk expression was considered beneficial. A quiet and peaceful place and having the child near were mentioned as crucial for milk expression. If the milk release was unsuccessful, they supported the mothers, for example by reassuring them that every little drop of milk was important and that they were doing a fantastic job, suggesting increased resting time, contributing to proper eating and drinking and encouraging increased frequency of stimulation. For those mothers who struggled with milk expression, it was important to give them time to be sad and disappointed and to clarify that it is natural and common to feel such emotions.

Nurses found that it could be difficult to follow evidenced-based practice. One nurse gave the example of forcing the mother’s nipple into the infant’s mouth; she knew that was not the best practice, but sometimes found it difficult to comply. Use of a nipple shield or a pacifier and weighing the infant before and after breastfeeding were described as both supportive and unsupportive for breastfeeding, depending on the context. Bottle-feeding was another issue that was mentioned, illustrating different ways of thinking. Some nurses found it frustrating that bottle-feeding was sometimes introduced too early, instead of waiting for the infant to mature and be able to suckle. Other nurses could see possibilities with bottle-feeding like earlier discharge. Nurses believed that the support given was influenced by their own experiences. Those who did not have their own experience of breastfeeding could see that as a shortcoming, while those who had their own experience of breastfeeding could find it difficult to set that aside and not allow their own experience to interact with the professional support provided. “It is possible to fully breastfeed premature infants, I know it! [laughing] I did it.” (Respondent B)

They talked about the need to incorporate the mother’s experiences, perceptions and previous advice into a nursing care plan. This was good for supporting the feeding process, but it could also clarify that nurses sometimes had difficulties adhering to and following the plan. “Then it is maybe up to us to follow what has been decided in the care plan. If I decide in the morning that we should try breastfeeding exclusively for 24 h, that it is done then, that the afternoon team doesn’t come in and change it too quick so we can’t evaluate it correctly.” (Respondent D)

Several of the nurses described supporting mothers as a complex and difficult task. Sometimes they had the impression that their efforts to be supportive had the opposite effect.

### 3.2. Seeing the Family as a Team

#### 3.2.1. The Preterm Infant as Leader of the Team

The nurses saw the preterm infant as an active part of the feeding process and highlighted that the feeding should be directed by the infant’s needs, in other words that oral feeding should be initiated based on the infant’s cues rather than on predefined times. Their experiences were that if the infant could lead the way, they usually preferred smaller amounts more frequently compared to the prescribed process, which was more often larger portions less frequently. They also emphasized that feeding was more likely to be successful if the infant’s competences were considered. “I find it so amazing with these preemies who have been gavage fed for so long and still they show these obvious cues that they have the instinct and willingness to suckle. And that they need to.” (Respondent F)

The infant’s feeding maturation was described as a process of competence, ranging from being skin-to-skin to being able to suckle. Initially, the infant’s cues were very discrete, but could for instance be noticed when the infant’s hands were placed near the mouth. In line with maturation, the infant become more active near the breast and started to lick, open the mouth and eventually suckle. A prerequisite for handling larger portions of breastmilk was being able to coordinate the competences of breathing, sucking and swallowing. Eventually, when the parents had learned to read the infant’s feeding cues and wake states, the cues guided the parents’ actions and the nurses took a step back and became supporters. The parents’ adherence to the infant’s cues were crucial for a successful feeding process.

#### 3.2.2. Roles of Other Family Members

The nurses believed that the feeding situation was something between the infant and the parents and something that could be very pleasant when it worked. They described the family as a unit, in which each member had a unique task related to the feeding. The mother’s role was to express milk and practice breastfeeding, while the father’s role could be to tube feed the infant and support the mother emotionally and practically. Their perceptions were that the parents’ involvement in the feeding had a positive impact on the family unit.

The nurses’ experiences were that if the mother had a relaxed and positive attitude, this was beneficial for breastmilk expression, as were the father’s views of breastmilk and breastfeeding. They found the father’s presence and support important for the mother to continue to maintain breastmilk production, especially when the mother only produced small amounts of breastmilk and was emotionally affected by this.

The closeness between parent and infant was important in order to create a relationship and to strengthen the mother’s ability to breastfeed. Therefore, the nurses worked to encourage the parents and underpin their closeness to their infants. In the same way as closeness was a prerequisite for the feeding process, separation between parents and the infant was described as a hindering factor for a successful feeding process.

The nurses found it important to be available for the families in case they needed them, but also to adapt their interference and support according to the needs of the family. In the following quotation, one of the nurses explained how she usually informed the parents: “I say, we are always here. We don’t expect you to know how to do this [feeding the infant], and if you are unsure, we will help. But it is your baby, and the baby should feel that it is the mother or the father who feeds them because that is how it would have been if you were at home…” (Respondent C).

The nurses also mentioned siblings. The nurses’ perceptions were that mothers sometimes seemed split between whether they should attend to the newborn infant or the sibling, which could have a negative impact on the feeding process. The importance of being aware of the needs of the whole family was highlighted. “Parents are so divided when they end up here [in the NICU] and if they have other children at home too, it is a catastrophe I would say… they are in the wrong place all the time… It is a huge feeling of calm and security to have them [the other children] here, yes, and to have the hubby around to support them when they feel down, and vice versa.” (Respondent B)

However, the nurses also described how such an early delivery could obstruct and delay a mother’s ability to feed the infant independently and according to own beliefs, compared to when the infant was born full-term and how that in turn could affect the parental role. Since motherhood is often closely related to breastfeeding, there is a risk that mothers will feel a loss of their role. The nurses described how they tried to empower and support mothers in their roles by emphasizing that they were good mothers, regardless of whether or not they were breastfeeding.

## 4. Discussion

The nurses experienced the oral feeding process when the infant is born extremely or very preterm as a complex and enduring collaboration. The unique process due to the unexpected early birth brought aggravating feeding factors to the already vulnerable mother-infant dyad. Poor medical conditions for the mother, anxiety and worry and low self-efficacy and self-confidence were mentioned by the nurses as hindering factors in this process. They described efforts to empower the mothers by providing practical and emotional support, which is in line with previous studies [14,25]. Still, mothers wish for more emotional support from nurses [15]. Since it might be clinically unfeasible to support all mothers equally, it is important to identify the mothers who are more vulnerable and at risk early on, in order to make appropriate use of resources. The globally applied Breastfeeding Self-Efficacy Scale can be used to score mothers’ self-efficacy [26]. However, based on the nurses’ responses in the present study, it might also be necessary to measure other variables such as self-confidence [20].

In FC neonatal care when parents are present and provide care at all hours of the day, parents need to develop their competence and understand the different parental role that comes with giving birth to a preterm infant [5]. In this study, the nurses had an intention to contribute to the parents’ understanding of the extended and individual feeding process. They wanted to support, inform and educate the family, but at the same time give the parents space and autonomy to support and take care of each other. In many ways, they saw themselves as guides or supporters who facilitated the feeding process, providing more support at the beginning and less later on when the parents had learned to understand the infant’s feeding cues. However, this is a balancing act and it might sometimes be misinterpreted by parents when the nurses do not give direct, practical support. In previous studies, it was found that the most frequently given support was related to breastmilk expression [14], while mothers in another study from an FC NICU wished for more support with breastmilk expression and breastfeeding practice [15]. Fathers have described that they find it difficult to provide support with milk expression, but also that the nurses—in their effort to provide support—sometimes asked the mothers too often if they had expressed milk [16]. Providing breastmilk and breastfeeding support is an important and delicate task for nurses in the NICU that needs to be customized and delivered individually. Some parents might need more practical support than others. Nurses’ intentions to provide autonomy and space for the family might be misinterpreted and lead to misunderstandings and feelings of a lack of support. It might be that within FC neonatal care, nurses should be more proactive and ensure that the family is not wandering through the unknown, searching for support on the internet as described among mothers of full-term infants in the UK [27]. Moreover, instead of asking about the frequency of pumping, parents can easily use a lactation diary to document expression times and milk amounts on paper or online for nurses to view [28].

Corresponding to the idea of FC neonatal care [5], the nurses valued the family as a team in which both parents and the premature infant are important for a successful feeding process. The infant is clearly at the center of the team and the one who directs the actions. However, this demands good infant-parent communication. In line with the BFHI 10 steps [9], the nurses found it important to guide the parents in understanding and reading the infant’s cues in order to ensure good communication within the team. Since each infant is an individual person, nurses need to be skilled at reading and interpreting preterm infants’ cues, in order to guide the parents in the right direction. There are programs that provide such training for parents and/or healthcare professionals, such as the Neonatal Behavioral Assessment Scale [29], the Neonatal Individualized Developmental Care and Assessment Program [5] and the Early Collaborative Intervention [30]. However, education and experience are also important [11], especially when it comes to feeding cues and not least listening to the parents and empowering them to trust their own parental skills and competences.

When talking about breastfeeding and milk expression, nurses mention fathers as a necessary practical and emotional source of support for the mothers, but do not say anything about what kind of support fathers need in order to handle the situation and provide support for mothers. A previous study shows that fathers in the NICU can be supportive of mothers [31], but less is mentioned about specific support for fathers regarding the feeding process of a preterm infant in the NICU. This warrants more research on how nurses can also support fathers’ emotions and roles in the feeding process in an FC NICU and, moreover, how that in turn impacts the feeding process for the preterm infant.

The strengths of this study are the in-depth qualitative interviews and the rigorous process to explore nurses’ experiences of the feeding process when the infant is born extremely or very preterm and cared for in an FC NICU. The included respondents represent a variety of experience of neonatal care. Some limitations should be noted. The participating wards were family-centered, small NICUs where parents can stay around the clock and be part of the care given to the infants. This is characteristic for Swedish neonatal units but may be different in other countries, thus limiting transferability. Moreover, some time has passed since the interviews were conducted, which must be taken into account. However, there is nothing in the transcripts implying that the FC neonatal care was different compared to today. The findings in this study have also been validated and verified by nurses currently working in the participating NICUs.

## 5. Implications for Practice

It is important to reflect on how information concerning nutrition and feeding is delivered and communicated in the FC NICU. A combination of written and verbal communication should be emphasized to avoid misunderstandings in the feeding process and nursing care plans can ensure an individual care. To make the best use of resources, nurses could screen parents to identify the families who need the most support and provide emotional and practical feeding support according to their individual needs.

## 6. Conclusions

Nurses’ experiences of the feeding process for extremely and very preterm infants in the FC NICU are described as a complex and long-lasting collaboration with vulnerable families. They saw the family as a team with all members involved in the feeding process, which corresponds to the concept of an FC NICU. The preterm infant was the leader of the team, as his or her needs and developmental stages directed the feeding. The nurses’ intentions were to guide and support parents in reading the infant’s feeding cues so that parents could eventually provide the care independently within their family team. They confirmed that breastfeeding support is a difficult task and acknowledged that their efforts to be supportive sometimes had the opposite effect.

## Figures and Tables

**Table 1 children-09-00016-t001:** Sub-categories, generic categories and main category.

Sub-Categories	Generic Categories	Main Category
A vulnerable process. Contributing to an awareness of a different feeding process. Providing professional support.	Guiding through the feeding process	A complex and long-lasting collaboration
The preterm infant as leader of the team. Roles of other family members.	Seeing the family as a team	

## Data Availability

The data presented in this study are available in Swedish on request from the corresponding author. The data are not publicly available due to the risk of identification.

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
