# Peer review of "Supporting Premature Infants’ Oral Feeding in the NICU—A Qualitative Study of Nurses’ Perspectives"

_children, 2021, doi:10.3390/children9010016_

Round 1
Reviewer 1 Report
Dear authors, first of all I would like to thank you for your research efforts.
Here are some notes for improvement
Introduction.
It is correct, although I see that they make some assertions that are not supported by any citations. Please correct this section by adding the omitted bibliographic
Participants
Explain the experience of the participants... years of experience and what type of experience.
Data collection.
Explain how you tested the questions
results
ok
Discussion
ok
Conclusions
This section should be redone, I think its conclusions are a bit poor. Focus the conclusions on the objectives and draw them from the results.
Good article, congratulations
Reviewer 2 Report
- The article emphasizes the role of nurses in the care of lactation and support for breastfeeding premature babies. What is the situation in the context of lactation consultants? Do women have access to the help they provide?
- Please explain the reason for publishing such old data. In my opinion, 6-7 years is a long enough period of time, especially when we take into account not only the recommendations but also the behavior of people, in order to observe significant changes in many issues. Over such a long period of time, changes in the perception of certain issues by society or changes in the health care system may occur. I understand that authors have verified the data, however, it is only a verification and not the current state of the situation faced by parents of premature children. I think it is worth describing in more detail the statements of the nurses participating in the verification of the current state.
- Is feeding according to the clock still used in the care of premature babies?
Author Response
- The article emphasizes the role of nurses in the care of lactation and support for breastfeeding premature babies. What is the situation in the context of lactation consultants? Do women have access to the help they provide?
Author Response: Yes, all women in the neonatal intensive care unit have access to support from nurses experienced and educated in the area. But there are no healthcare staff that provide lactation support only i.e lactations consultants. Nurses usually care for one to a few families at a time, depending on the care level, and provide all nursing care for these families. We have added education levels of the participants in section 2.3.
- Please explain the reason for publishing such old data. In my opinion, 6-7 years is a long enough period of time, especially when we take into account not only the recommendations but also the behavior of people, in order to observe significant changes in many issues. Over such a long period of time, changes in the perception of certain issues by society or changes in the health care system may occur. I understand that authors have verified the data, however, it is only a verification and not the current state of the situation faced by parents of premature children. I think it is worth describing in more detail the statements of the nurses participating in the verification of the current state.
Author Response: Yes, it is some years ago since the data was collected. The delay was initially because the PhD-student working with the project had to withdraw due to a sick son. The project was further delayed because of a heavy workload, a change of University (and country) for me as a lead on the project, and a need for prioritizing research projects with external grants. This study had no external funding. This is the third article in a larger study where we have been investigating experiences of oral feeding of very and extremely preterm infants in the neonatal intensive care unit from the perspectives of mothers, fathers, and nurses respectively. One explanation for a delayed submission of this manuscript is our intention to publish them one at a time and all journals are not providing as rapid review processes as MDPI Children. Article 1 was published in 2020 and number 2 in 2021. We used data source triangulation in article 2 as well to increase trustworthiness. We have now included a reference to that article in section 2.7. We have also included this as a limitation to the study in the end of the discussion. Moreover, a quote from one of the nurses participating in the triangulation: “…we base our work on the child being a competent individual, we try to see for ourselves, and help the parents’ to see the child’s cues in order to establish a feeding situation that works.” Since this is the first time the subject of oral feeding for extremely preterm infants has been explored in family-centered neonatal units in this way, we find it important to share the findings with colleagues despite the data is a bit old. To our knowledge, MDPI Children has no policy regarding old data sets.
- Is feeding according to the clock still used in the care of premature babies?
Author Response: Yes, if the baby is extremely preterm feeding according to the clock may be provided for a rather long period of the baby’s postnatal age. Sometimes it is also necessary to provide continuous feeding. This is because of the infant’s immaturity; they tolerate only small amounts (e.g. 1 mL) and they don’t have the strength to show hunger cues. This changes by day, and as soon as the baby is more mature, she/he can start with larger portions, longer time intervals, and also to decide when to be fed.
Reviewer 3 Report
Mӧrelius and colleagues have done a remarkable work on the nurse's perception on supporting premature infants' oral feeding in NICU in Sweden.The manuscript is worth publication following revision as per the comments outlined below:
Introduction:
This section is very brief, the reader may require a concise elaboration on the topic. The authors are advised to briefly elaborate the rationale of the study and current context.
Methods:
Study design should be described in detail rather than just a single phrase. what kind of qualitative study? any theoretical framework? The study design is lacks adequate informative.
Was data saturation was attained by 9 participants? how was this confirmed?
The information in section "Trustworthiness" is well written. However, in order to enhance the rigor of the study, were there any approaches undertaken?
The authors are recommended to refer to the Consolidated criteria for reporting qualitative research (COREQ) checklist.
Results:
The results section is well presented, with substantial number of themes, subthemes, and direct quotes of the participants. The findings of the study are interesting as well as novel.
Discussion:
The discussion is well-written and provides an interesting interpretation of the findings of the study.
The limitations of the study are not well explored. The authors are suggested to highlight the limitations of the study in this section as well.
The authors are also recommended to add a sub-section highlighting and focussing the implications and recommendations drawn from this study.
Round 2
Reviewer 2 Report
I accept the authors' responses to my comments.